# UNITENTION: ATTEND A SAMPLE TO THE DATASET

## ABSTRACT

We propose a trainable module termed **Unitention**, an abbreviation for **un**iversal-**in**dividual cross-at**tention**, to improve deep features of a given neural network by attending the feature of a data sample to those of the entire dataset. This innovation is motivated by two key observations: (i) traditional visual encoding methods, such as Bag of visual Words, encode an image by using a *universal* dataset-wide codebook, while (ii) deep models typically process every *individual* data sample in isolation, without explicitly using any *universal* information. Our Unitention can bridge this gap by attentively merging universal and individual features, thus complementing and enhancing the given deep model. We evaluate its efficacy on various classification benchmarks and model architectures. On ImageNet, Unitention improves the accuracy of different ConvNets and Transformers. In particular, some $k$-NN classifiers with Unitention can even outperform baseline classifiers. Improvements in fine-grained tasks are more substantial (up to $2.3\%$). Further validations on other modalities also confirm Unitention's versatility. In summary, Unitention reveals the potential of using dataset-level information to enhance deep features. It opens up a new backbone-independent direction for improving neural networks, orthogonal to the mainstream research on backbone architecture design.

## 1 INTRODUCTION

The remarkable capabilities of contemporary deep learning methods have been prominently demonstrate in different domains, ranging from image (Chen et al., 2020; Bao et al., 2021; He et al., 2021; Tian et al., 2022), language (Devlin et al., 2018; Brown et al., 2020; Dong et al., 2019), signal processing (Cheng et al., 2023a; Dong et al., 2018; Cheng et al., 2023b;c) to machine intelligence (Silver et al., 2016; 2017; Vinyals et al., 2019). The explosion in deep model architecture is considered to be one of the most driving forces and lies in the center of deep learning research (He et al., 2016; Vaswani et al., 2017). Since the advent of AlexNet (Krizhevsky et al., 2012), which rekindled widespread interest in neural networks, efforts have been made to explore more powerful and efficient neural architectures, either by hand design (Hu et al., 2018; Howard et al., 2017; Sandler et al., 2018) or automated search (Guo et al., 2020b; Yu et al., 2020; Guo et al., 2020a; Cai et al., 2019). Many landmark network families have been proposed, such as ResNets (He et al., 2016) and Transformers (Vaswani et al., 2017), dominating different application areas (vision or language). For more advanced architectures, some have focused on merging the advantages between these two families and have made significant progress (Dosovitskiy et al., 2020; Wang et al., 2021; Yuan et al., 2021; Wu et al., 2021; Liu et al., 2021; Yu et al., 2022; Liu et al., 2022a).

Beyond the vast amount of mainstream research on backbone architectures, *one might ask*: *could there exist an architecture-independent approach capable of enhancing deep model performance?* To explore this, we looked back to the classics of visual representation prior to deep learning. The landscape of visual representation was once dominated by techniques such as Bag of visual Words (BoW, Csurka et al. (2004); Perronnin et al. (2006); Van Gemert et al. (2009)) and Vector of Locally Aggregated Descriptors (VLAD, Jégou et al. (2010); Sánchez et al. (2013)). A pivotal component in these methodologies was a universal codebook that characterizes the distribution of entire dataset. Its construction involved the normalization and clustering of diverse visual descriptors extracted from training images within the dataset. During inference, a test image was represented as cluster assignments (Csurka et al., 2004; Perronnin et al., 2006; Van Gemert et al., 2009) or accumulated "distances" to the nearest codewords in the universal codebook (Jégou et al., 2010; Sánchez et al., 2013). We illustrate this process as **Universal Encoding** in figure 1 (a).

*How to extract the feature of a data sample x:*

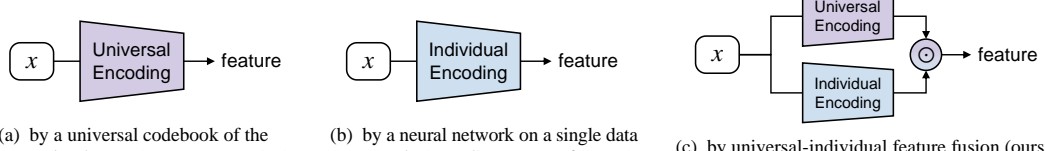

(a) by a universal codebook of the entire dataset (*e.g.*, BoW, VLAD)

(b) by a neural network on a single data sample (*e.g.*, CNNs, Transformers)

(c) by universal-individual feature fusion (ours)

Figure 1: Schematic diagrams of how to encode a data sample. "⊙" in (c) denotes an arbitrary fusion operator.

A key difference between deep models (He et al., 2016; Vaswani et al., 2017) and traditional BoW-based approaches (Csurka et al., 2004; Jégou et al., 2010) is that the former extract features solely from an input sample alone, without any explicit interaction with dataset-wide features like a universal codebook. We term this paradigm as **Individual Encoding**, and visualize it in figure 1 (b).

Given such a disparity between universal and individual encoding, a conjecture naturally emerges: an integration of these two different ways could complement and enhance each other, yielding a more robust and refined feature. This conceptual model manifests itself as a new **Universal-Individual Encoding** paradigm in figure 1 (c). We find this idea *exactly satisfies* the aforementioned need for an architecture-independent way to improve the capability of deep model: it could be applied to any neural network backbone (*e.g.*, ResNets and Transformers) without modifying their architectures.

However, this universal-individual fusion mechanism remains largely unexplored. As an initial exploration, we conduct an inference-only study to test whether the mechanism can make deep features more discriminative. The results, detailed in figure 2 and further discussed in section 2.1, show that a *training-free* module after the backbone can improve test performance by simply merging "universal" and "individual" features. This reveals the potential of universal-individual feature fusion.

Our goal now boils down to designing an *end-to-end trainable* module to facilitate this fusion. We explored several designs and came up with three key principles for the module: 1) maintaining a universal feature bank to represent the entire dataset; 2) applying cross-attention between the bank and the deep feature of current data sample (individual); 3) merging the cross-attended result as output. The resulting module, called **un**iversal-**i**ndividual cross-at**tention** or Unitention, serves as a general plug-and-play module that can be used on top of various backbone models.

We tested Unitention on several classification benchmarks across different data modalities involving 2D images and 1D signals, and observed consistent performance gains (up to 2.3%) over a variety of state-of-the-art models. Some $k$-NN classifiers based on Unitention-enhanced deep features even outperform those baseline classifiers, further demonstrating its effectiveness. All of these results demonstrate Unitention can enhance various powerful deep models. This universal-individual mechanism opens up a new backbone-independent direction for improving neural networks, orthogonal to the mainstream research on backbone architecture design and refinement. We hope that Unitention can serve as a generic plug-in for modern deep models, and would inspire more future work towards better exploitation of dataset-level information.

To summarize, the contributions are three-fold:

- We propose and explore the universal-individual fusion mechanism, providing a new backbone-independent direction to improve deep model's capability. This trajectory stands in contrast to existing research paradigms that mainly focus on the architecture design of backbones.
- We conduct a training-free case study to test this mechanism and demonstrate its potential. We further exploit its potential by developing an end-to-end trainable module (Unitention).
- We have done comprehensive experiments and observed Unitention can deliver consistent performance gains across a wide range of both 1) data modalities and 2) model architectures.

## 2 METHOD

We present an *training-free* case study in section 2.1 as a test of the validity of universal-individual feature fusion mechanism. The results directly motivate us to design an end-to-end *trainable* module called Unitention to facilitate such a mechanism, detailed next in section 2.2.

## 2.1 A Training-free Case Study

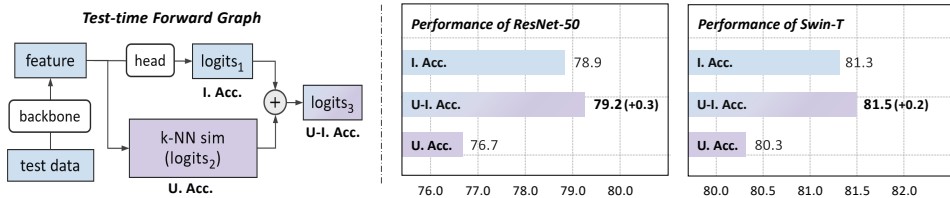

Figure 2: **A training-free study to test the proposed universal-individual fusion mechanism.** Two models are trained on ImageNet and then tested, each of which consists of a backbone and a linear layer (classification head). $\text{logits}_1$: prediction of the trained linear classification head; $\text{logits}_2$: prediction of a $k$-NN classifier after the backbone; $\text{logits}_3$: prediction of a training-free universal-individual fusion ($\text{logits}_3 = \text{logits}_1 + \text{logits}_2$). ImageNet validation accuracy is shown in the right. Higher is better. Improvements in accuracy are noted.

**Setting.** We take classification models that have already been supervised trained on ImageNet (Deng et al., 2009) for investigation. Each classification model consists of a neural network backbone and a linear classification head. Here, we use two representative backbones: ResNet-50 (He et al., 2016) and Swin-Tiny (Liu et al., 2021). The basic idea of this study is to design a training-free module to implement a universal-individual feature fusion, and to check if it could improve deep model's test accuracy. We are going to assess the module by comparing the performance in three different evaluation settings: individual-only, universal-only, and universal-individual.

**Evaluation.** We start by specifying what the "individual-only" evaluation is. Consider the plain model evaluation that directly uses its own internal linear classifier. It is actually the individual-only way, as the classifier make predictions based only on a single data point's feature. This individual-only accuracies of original ResNet-50 and Swin-Tiny are illustrated in figure 2 as "I. Acc.".

Recall the universal feature is usually obtained by some dataset-level structure like the codebook in BoW/VLAD. We find the simple $k$-nearest neighbors ($k$-NN) algorithm is a special case of "universal-only" approach, since $k$-NN similarities between a sample and all its neighbors could be regarded as the dataset-level (universal) information. We thus calculate the $k$-NN accuracies based on the representations after backbones (omitting the trained linear classifiers), and show them as "U. Acc." in figure 2.

Now we need to design a module for universal-individual feature fusion. A natural idea is to mix the $k$-NN similarities with the individual prediction. To achieve this, we normalize $k$-NN similarities to log-probabilities ($\text{logits}_2$) and then add them with linear classifier's logits ($\text{logits}_1$) to get the fused ones ($\text{logits}_3 = \text{logits}_2 + \text{logits}_1$). More details (source codes) can be found in Appendix A. By applying probability function (e.g., $\text{softmax}$) on $\text{logits}_3$, we get the so-called "universal-individual" predictions. The resulted accuracies are reported in figure 2 as "U-I Acc."

**Observation and motivation.** From figure 2, one can observe both ResNet-50 and Swin-Tiny enjoy this universal-individual feature fusion and achieve higher accuracies without any further training. It verifies $k$-NN similarities here would complement and enhance the linear classifier, thus improves the whole model's representation. In this case, considering universal information and individual information simultaneously could give a better representation than considering any of them alone, even if no extra *training* is conducted. This motivates us to develop a *trainable* module to better explot its potential.

## 2.2 An End-to-end Trainable Module: Unitention

**Overall designs.** Recall the $k$-NN experiments in section 2.1, where a $k$-NN classifier is put on top of a trained backbone and predicts image labels. To some extent the $k$-NN evaluation is like a retrieval process on the whole dataset, seeking for some similar samples and then making the prediction. Likewise, we plan to design a module act as a role that "retrieves" dataset, finds representative samples, and then fuses their features with the current to achieve universal-individual fusion. Thanks to the powerful attention mechanism (Vaswani et al., 2017), we find cross-attention operator (Vaswani et al., 2017) could be a natural choice to achieve the retrieval and fusion. Specifically, we

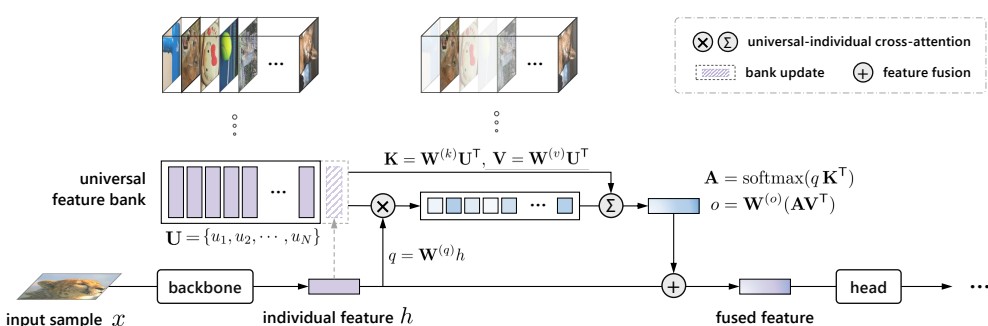

Figure 3: **An overview of Unitention module,** which is inserted between the backbone model and its classification head, taking deep feature $h$ as input, applying universal-individual fusion, and finally yielding a fused feature. The universal feature bank $\mathbf{U}$ stores historical features $h$ as the dataset-level information. The backbone, Unitention module, and linear classification layer will be trained together in an end-to-end manner.

are designing this Unitention module following the rules:

- Unitention takes the backbone's output feature vector as its input.
- A universal feature bank needs to be maintained to offer dataset-level feature during training.
- The cross-attention is applied for universal-individual feature fusion. The only query is from the individual feature, and multiple keys and values are derived from that universal bank.
- The attention result should finally be merged with the individual feature to enhance it.

An overview of Unitention is illustrated in figure 3. We would detail it as follows.

**Module input and universal feature bank.** Let $x$ denote a data point fed into the backbone. Its deep feature vector $h$ is the input of Unitention module, which would be used to update a universal feature bank. As a feature set representing the whole dataset, this bank contains $N$ feature vectors $\{u_1, u_2, \cdots, u_N\}$, packed together into a matrix $\mathbf{U}$. One can use a first-in-first-out (FIFO) queue, momentum-updated class centers, or any other datastructures to maintain $\mathbf{U}$. We leave the practical implementation in section 3.1. In summary, this feature bank will provide universal information, and is updated by every single feature vector $h$ during the training.

**Cross-attention between bank and individual.** Without loss of generality, we consider the *single*-head attention mechanism here. We draw on the standard formulation of cross-attention from Vaswani et al. (2017). However, the query $q$ here is not a sequence of vectors, but a unique vector, as $q$ is based on the only feature vector $h$. The query is calculated by:

$$q = \mathbf{W}^{(q)} h, \tag{1}$$

where $\mathbf{W}^{(q)}$ is the query transform matrix. A key set $\{k_1, k_2, \cdots, k_N\}$ and a value set $\{v_1, v_2, \cdots, v_N\}$ are obtained based on the universal bank and formed into matrices $\mathbf{K}$ and $\mathbf{V}$:

$$\mathbf{K} = \mathbf{W}^{(k)} \mathbf{U}^{\mathsf{T}}, \tag{2}$$

$$\mathbf{V} = \mathbf{W}^{(v)} \mathbf{U}^{\mathsf{T}}. \tag{3}$$

The "universal-individual" cross-attention is calculated between the unique query and $N$ key-values:

$$\mathbf{A} = \text{softmax}(q \mathbf{K}^{\mathsf{T}}), \tag{4}$$

$$o = \mathbf{W}^{(o)}(\mathbf{A}\mathbf{V}^{\mathsf{T}}), \tag{5}$$

where $\mathbf{W}^{(o)}$ is projection weight and $o$ the result. $\mathbf{A}$ is a row-vector attention map shaped in $1 \times N$ rather than a two-dimensional matrix, as it shows the affinity between the only query and $N$ keys.

We highlight a key difference between the attention here and standard cross-attention: In the standard, queries and key-value pairs are from two difference feature sequences of the *one same* input data point. While in Unitention, the query is from one data point $x$ but keys and values are based on the universal feature bank that contains rich information from many other data points. Therefore, Unitention supports such an inter-sample feature interaction, which otherwise is not possible in standard cross-attention.

**Feature fusion and module output.** Given the hypothesis that the universal-individual attention result $o$ could enhance the deep representation $h$, we would merge them in some way. This could be achieved by simple element-wise addition, multiplication, or even more sophisticated operations. We would discuss the choice further in section 3.1 and section 3.6.

## 3 EXPERIMENTS

### 3.1 IMPLEMENTATION DETAILS

**Overall Implementation.** A classification neural network model typically consists of a backbone (*e.g.*, ResNet) and a classifier head (usually a linear layer). We insert the Unitention module between backbone and classifier, and then train these three part in an end-to-end manner. Details about Unitention are introduced as follows.

**Universal bank.** The universal bank is a set of feature vectors $\{u_1, u_2, \cdots, u_N\}$ which represents the whole dataset. We use online class centers as the bank in all experiments unless otherwise specified. In this way, bank size $N$ is the number of classes in dataset. To update the bank, we use every deep feature $h$ and its label $y \in \{1, 2, \cdots, N\}$ to update the $y$-th class center representation:

$$u'_y \leftarrow \alpha u_y + (1 - \alpha) h, \tag{6}$$

where $\alpha$ is a moving average ratio. We found a relative small $\alpha$ (*e.g.*, 0.9) ensures timely updates, thus providing fresh universal features and achieving better results than larger ones like 0.95. We use 0.9 for all experiments below and provide an ablation in section 3.6.

**Cross-attention.** We use the standard cross-attention architecture in Vaswani et al. (2017). This block contains a standard multi-head attention module, a two-layer perceptron, and two layer normalizations before them. The hyperparameters of multi-head attention are detailed in Appendix A. Following Bao et al. (2021), we use the layer scale (Touvron et al., 2021) as we found it can stabilize the training. We also reported the efficiency evaluation in Appendix A.

**Feature fusion.** As mentioned in section 2.2, there are several choices to fuse attention output $o$ with the individual feature $h$. Compared to element-wise multiplication like in Hu et al. (2018), we found adding the features up is good enough. See section 3.6 for more details.

### 3.2 RESULTS ON IMAGENET CLASSIFICATION

**Setting.** We first evaluate Unitention on the general image classification benchmark ImageNet Deng et al. (2009), a challenging large-scale dataset with 1,000 classes. Its training split and test split contain about 1.28 millions and 50 thousands images respectively. We use two mainstream backbones in computer vision, namely convolutional netowrks (convnets) and vision transformers (ViTs), to test Unitention. We thoroughly use a wide range of models in this study, including convention backbones (ResNet He et al. (2016), ResNeXt Xie et al. (2017), SE-Net Hu et al. (2018), ViT Dosovitskiy et al. (2020)) and advanced architectures (ECA-ResNet Wang et al. (2020a), Reg-Net Radosavovic et al. (2020), VAN Guo et al. (2022), Swin Transformer Liu et al. (2021)). See Appendix C for more details on ImageNet experiments. The full model that contains Unitention module, backbone and linear classifier head, is trained from scratch in an end-to-end manner and then tested on test data split. Here we consider two evaluation metrics: standard accuracy and $k$-NN accuracy. The standard accuracy is based on the predictions made by the full model. While in $k$-NN evaluation, the trained classification head is omitted, and the intermediate features are used. This $k$-NN metric is intended as a complement by directly measuring the linear separability of features.

**Standard Performance (with model's heads).** The standard accuracy of models with and without Unitention are listed in table 1. "BL" represents the baseline performance directly quoted from the original papers. For a fair comparison, we re-implement all baseline models and report their updated baseline accuracy as "BL*". "Ours" denotes model performance with Unitention. The absolute accuracy improvements from Unitention are also reported. Comparing BL* results with Ours, one can observe steady accuracy gains brought by Unitention. It verifies Unitention can consistently

Table 1: **Main results on ImageNet.** "BL": **B**ase**L**ines quoted directly from original papers; "BL*": our re-implementations; "Ours": performance with Unitention. Absolute improvements over re-implemented baselines are noted after our performance. The last column shows whether a $k$-NN classifier with Unitention outperforms the baseline model, *i.e.*, "our $k$-NN Acc." is higher than "BL* Standard Acc.".

| Backbone | Standard Acc. | | | $k$-NN Acc. | | $k$-NN better |
|---|---|---|---|---|---|---|
| | BL | BL* | Ours | BL* | Ours | |
| *ConvNets.* | | | | | | |
| ResNet-50 | 75.30 | 78.92 | 79.90 (+0.98) | 76.71 | 79.21 (+2.50) | ✓ |
| ResNet-101 | 76.40 | 80.25 | 80.97 (+0.72) | 78.81 | 80.19 (+1.38) | ✗ |
| ResNet-152 | 77.00 | 81.03 | 81.64 (+0.61) | 79.68 | 80.60 (+0.92) | ✗ |
| ResNeXt-50-32x4d | 77.80 | 79.23 | 80.01 (+0.78) | 78.66 | 79.32 (+0.66) | ✓ |
| SE-ResNet-50 | 76.71 | 79.45 | 80.14 (+0.69) | 78.07 | 79.25 (+1.18) | ✗ |
| ECA-ResNet-50-T | 77.48 | 80.11 | 80.62 (+0.51) | 78.98 | 79.75 (+0.77) | ✗ |
| RegNetY-16GF | 80.40 | 82.16 | 82.87 (+0.71) | 81.30 | 81.87 (+0.57) | ✗ |
| VAN-Tiny | 75.40 | 75.84 | 76.94 (+1.10) | 75.44 | 76.56 (+1.12) | ✓ |
| VAN-Small | 81.10 | 81.11 | 81.63 (+0.52) | 80.52 | 81.26 (+0.74) | ✓ |
| *Transformers.* | | | | | | |
| ViT-Small | 79.9 | 81.27 | 81.85 (+0.58) | 80.34 | 80.78 (+0.44) | ✗ |
| ViT-Base | 81.8 | 81.84 | 82.20 (+0.36) | 80.91 | 81.21 (+0.30) | ✗ |
| Swin-Tiny | 81.3 | 81.28 | 81.95 (+0.67) | 80.34 | 80.62 (+0.28) | ✗ |

boost the performance across both convention and advanced backbones. Notably, significant improvements over strong baselines (VAN and Swin) indicate even the most powerful architectures to date can still benefit from Unitention. Besides, we find Unitention improves the accuracy of SE-ResNet (a channel-wise attention network). This shows that the "dataset-wise attention" could further complement the channel-attended features.

$K$**-NN Performance (omitting model's heads).** In this evaluation, all trained classification heads are removed. We directly use the intermediate features after backbone (BL* in table 1) or Unitention (Ours) to perfom $k$-NN -based classification. Comparing in the right panel of table 1, one can observe Unitention surpasses baselines by large margins (up to 2.5%). This verifies that Unitention does reinforce the deep features of baseline models by making them more *linear separable*. Note that some $k$-NN classifiers with Unitention can outperform the learned linear classifier within baseline models, *i.e.*, "our k-NN Acc." is higher than "BL* Standard Acc." (as shown in the last column in table 1). This further demonstrate the superiority of features learned by Unitention.

Table 2: **iNaturalists accuracy.** $\Delta$: difference between baseline (w/o Unitention) and ours (w/ Unitention).

| Dataset | iNaturalist 18. | | iNaturalist 19. | |
|---|---|---|---|---|
| Backbone | Swin-T | Swin-S | Swin-T | Swin-S |
| Baseline | 72.5 | 76.4 | 76.8 | 80.8 |
| Ours | 74.8 | 78.0 | 78.7 | 82.3 |
| $\Delta$ | +2.3 | +1.6 | +1.9 | +1.5 |

## 3.3 RESULTS ON FINE-GRAINED CLASSIFICATION

Next we assess Unitention's efficacy on two challenge fine-grained datasets. The iNaturalist 2018 dataset (iNat18) Van Horn et al. (2018) contains around 450,000 training and validation images from 8,000 natural fine-grained categories. On the other hand, the iNaturalist Challenge 2019 Van Horn et al. (2018) contains 1,010 species with a combined training and validation set of 268,243 images. These two long-tail and fine-grained datasets pose a high challenge to the recognition ability of the deep model. See Appendix D for more details on iNaturalist experiments. From table 2 we can see that Unitention significantly helps the deep model to overcome such challenges, as it brings more significant improvements compared to those on ImageNet. We attribute this success primarily to the

fact that dataset-level information is more useful for fine-grained tasks. This bodes well for the high potential application value of Unitention, as many real tasks are long-tailed and fine-grained.

Table 3: **Accuracy on signal classification benchmarks.** Improvements of Unitention are denoted as $\Delta$.

|  | | Classification Acc. | |
| --- | --- | --- | --- |
| Setting | BL | Ours | $\Delta$ |
| *Device signals* | | | |
| MLP | 66.4 | 67.5 | +1.1 |
| FCN-LSTM (Karim et al., 2017) | 76.7 | 77.6 | +0.9 |
| Transformer (Vaswani et al., 2017) | 80.1 | 81.4 | +1.3 |
| *Sensor signals* | | | |
| MLP | 57.1 | 57.8 | +0.7 |
| FCN-LSTM (Karim et al., 2017) | 66.0 | 67.5 | +1.5 |
| Transformer (Vaswani et al., 2017) | 69.9 | 71.0 | +1.1 |
| *Medical signals* | | | |
| MLP | 62.8 | 64.5 | +1.7 |
| FCN-LSTM (Karim et al., 2017) | 72.0 | 73.3 | +1.3 |
| Transformer (Vaswani et al., 2017) | 75.2 | 76.2 | +1.0 |

### 3.4 RESULTS ON OTHER MODALITIES

This section aims at exploring whether Unitention can be generalized to other data modalities. Since Unitention is model-independent, we expect it to achieve this. Specifically, we choose several popular benchmarks for one-dimensional signal classification to test Unitention. Following the convention of signal processing, we use three of mainstream backbones: multi-layer perceptron model (MLP), LSTM-FCN Karim et al. (2017), and transformer Vaswani et al. (2017). See Appendix E for more details on these tasks.

For a more comprehensive assessment of Unitention, we consider using three datasets that contains different 1D signals: **(i) Device signals.** The ElectricDevices Bagnall et al. (2017) dataset contains 8,926 training sampels and 7,711 test samples of 7 signal classes. These data were collected as part of government sponsored study called Powering the Nation. , containing electricity readings from 251 households. Each 1D signal is length 720. **(ii) Sensor signals.** We use UWaveGestureLibrary Liu et al. (2009) that contains 896 and 3,582 samples in training and test split of 8 classes. A set of eight simple gestures generated from accelerometers. The data consists of the X,Y,Z coordinates of gestures motion that is collected by sensor. The data length is 315. **(iii) Medical signals.** EOGHorizontalSignal Fang & Shinozaki (2018) is also used, which has the same number of samples (362) in its training and test subset. It is a 12-classification task. The data are electrooculography signal (EOG), which is the measurements of the potential between electrodes placed at points.

The results are listed in table 3. Observations that are consistent with those in section 3.2 and table 1 can be found: Unitention has consistent and non-marginal performance improvements over different signal modalities and models. This indicates that Unitention can serve more general signal processing areas beyond the image processing.

### 3.5 RESULTS ON MORE CHALLENGING DOWNSTREAM TASKS

The above implementation of Unitention uses feature vectors after the global average pooling operation (GAP), and is applicable to global-feature-based classification tasks. We also investigated whether Unitention can be generalized to other dense prediction tasks, including object detection, instance segmentation, and semantic segmentation. For the sake of diversity, we have tested two different families of models (CNNs and Transformers). Specifically, the Mask R-CNN ResNet50-FPN (He et al., 2017; 2016) on MS-COCO Lin et al. (2014), and the UperNet Swin-T Xiao et al. (2018); Liu et al. (2021) on ADE20k Zhou et al. (2017). The performance improvements are promising (up to $+1.5AP$), even better than those on ImageNet. We attribute this to the fact that in more challenging tasks, the model requires a more informative and rich dataset-level understandings, which can be provided by our Unitention. See Appendix F for the detailed performance and analysis.

Table 4: **Ablation study** on the influence on each key component in our method. [†] is the learned value of $\tau$.

|  | Description | Universal Bank | Fusion | Mom. | $\tau$ | Acc. | $\Delta$ |
|---|---|---|---|---|---|---|---|
| 1 | w/o Unitention | | | | | 78.9 | -1.0 |
| 2 | Unitention (default) | class centers (1000) | add | 0.9 | 1.0 | 79.9 | 0.0 |
| 3 | self-attention | individual feature (1000) | add | 0.9 | 1.0 | 78.6 | -1.3 |
| 4 | fewer class centers | class centers (100) | add | 0.9 | 1.0 | 79.2 | -0.7 |
| 5 | FIFO queue | FIFO queue (1000) | add | — | 1.0 | 79.3 | -0.6 |
| 6 | learnable centers | parameters (1000) | add | — | 1.0 | 79.4 | -0.5 |
| 7 | SE-style fusion | class centers (1000) | mul | 0.9 | 1.0 | 79.6 | -0.3 |
| 8 | $-$Momentum | class centers (1000) | add | 0.8 | 1.0 | 79.4 | -0.5 |
| 9 | $+$Momentum | class centers (1000) | add | 0.95 | 1.0 | 79.6 | -0.3 |
| 10 | $-\tau$ | class centers (1000) | add | 1.0 | 0.2 | 79.3 | -0.6 |
| 11 | $+\tau$ | class centers (1000) | add | 1.0 | 5.0 | 79.5 | -0.4 |
| 12 | learnable $\tau$ | class centers (1000) | add | 1.0 | 1.37[†] | 79.9 | 0.0 |

## 3.6 ABLATION STUDY

In this series of experiments, we try to understand Unitention better by studying the performance of its variants. All experiments are performed with a ResNet-50 on ImageNet (a 1000-classification dataset). Results are listed in table 4.

**Overall effectiveness of Unitention.** Comparing the performance of row 2 with row 3, it can also be concluded that the performance improvement of Unitention is not simply due to the increase in parameters, since row 2 and row 3 have exactly the same number of parameters and computations. This is another solid proof of Unitention's overall effectiveness.

**The importance of universal bank.** We find it is important to use all class centers. Using only 100 categories random selected from 1000 (row 4) hurt the performance. We also notice a simple FIFO queue does not work so well (row 5). The main reason could be that a queue of size 1000 is not sufficient for representing the entire dataset. And using larger sizes would be too inefficient. In addition, we also test Unitention with a learnable "class center" whose feature bank is replaced by trainable parameters updated by back-propagation (row 6). It also performs slightly worse than ordinary class centers. Taken together, the class centers (row 2) become our best choice.

**Fusion operation.** We observe simply adding up individual and universal features is good enough, since an SE-module style fusion in row 7 (element-wise multiplication) won't show any gains.

**Hyperparameters.** We then adjust the two most important hyperparameters in Unitention: the momentum of class centers, and the temperature $\tau$ of softmax in cross-attention. The best choices are 0.9 and 1.0, respectively. We also observe that a learnable $\tau$ is not necessary (as in row 12).

## 4 RELATED WORK

### 4.1 CLASSIC CODEBOOK-BASED VISUAL REPRESENTATION

Visual-word encoding, popularized by BoW (Bag of visual Words, Csurka et al. (2004)), had its heyday before the explosion of deep learning. A feature codebook for building dataset-level (universal) understanding is largely credited with its success. Typically, this codebook consists a large number of local descriptors like SIFT (Lowe, 1999; Bay et al., 2006; Rublee et al., 2011) descriptors, and is constructed in a fully unsupervised manner. It can also be regarded as a "visual vocabulary". An image can be encoded as frequencies of codewords, say, a histogram of codeword occurrences. This corresponds to a hard assignment. There have been many variants (Van Gemert et al., 2009; Farquhar et al., 2005; Perronnin et al., 2006; Boureau et al., 2010) to enhance the canonical BoW method, including VLAD (Vector of Locally Aggregated Descriptors, Jégou et al. (2010)) and FV (Fisher Vector, Sánchez et al. (2013)). Some focuses on soft assignments (Van Gemert et al., 2009;

Farquhar et al., 2005; Perronnin et al., 2006) or advanced patch aggregation mechanism (Sánchez et al., 2013; Jégou et al., 2010). FV (Sánchez et al., 2013) uses Fisher Kernel (FK) principle to reduce the codebook size as well as obtain stronger representations than BoW. VLAD (Jégou et al., 2010) considers distances between a given sample and its nearest neighbors, which is proved to be a simplified version of FV under some approximations (Jégou et al., 2011).

## 4.2 DEEP MODELS AND IMPROVED HEAD DESIGNS

Contrary to classic codebook-based algorithms, typical **deep models** extract features based only on the current single sample point without any explicit use of dataset-level information. For example, a classifier model can be viewed as a univariate vector function $f_\theta(\cdot) : \mathbb{R}^D \mapsto \mathbb{R}^C$ parameterized by $\theta$, whose input is a single data point ($D$-dim) and whose output is a vector of probabilities for each category ($C$ categories) to which it belongs. In computer vision, convolutional networks (Krizhevsky et al., 2012; He et al., 2016; Liu et al., 2022b) and vision transformers (Dosovitskiy et al., 2020; Liu et al., 2021) are two representative model families.

Besides these neural backbones, there have been many studies focusing on **boosting the heads**. To the best of our knowledge, these works are also based on a single data input. Model fusion like bilinear pooling Gao et al. (2016) merges features from different models on one single data. Xie et al. (2021); Zhu & Wu (2021); Wang et al. (2020b) enhance the classifier by replacing standard global average or max pooling with advanced global pooling methods. There are also some work (Zhu et al., 2017; Li et al., 2022) utilizes self-attention-like mechanism on one single data's deep feature along the spatial or channel axis. In sum, these single-data-based methods have been demonstrated helpful for classification heads.

## 4.3 DIFFERENCE BETWEEN CLASSIC REPRESENTATION AND DEEP MODELS

A huge difference between the classic visual models (Csurka et al., 2004; Jégou et al., 2010) and deep-learning based methods (He et al., 2016; Vaswani et al., 2017) of today is they are using radically different ways to encode images. BoW and its variants are originally training-free, and use a dataset-level codebook for encoding. The encoding process needs to consider the relationship between the current image and all historical ones in dataset. On the contrary, deep models forward every image independently to get a deep feature. They seem to understand the entire dataset in an implicit way. In this paper, we summarize this difference and hope to explore whether these two encoding approaches can complement and enhance each other.

## 5 LIMITATIONS

**More generalized universal bank.** The universal bank used currently has to be updated via classification labels. In the future, we will try to design universal banks that do not depend on labels, for example, by trying a FIFO queue with more prior knowledges. We believe a universal bank that can still be updated during the inference phase (on unlabeled test data) will make Unitention more comfortable with real-world challenges like out-of-distribution tasks, low-shot or few-shot settings.

## 6 CONCLUSION

Visual-word encoding like Bag of Visual Words had its heyday before the explosion of deep learning, thanks largely to a feature codebook for building dataset-level (universal) understanding. In contrast, modern deep neural networks forward each data sample (individual) independently, without any explicit universal modeling. Would deep models be enhanced by perhaps a universal-individual fusion mechanism? We conduct an inference-only case study to verify this, and develop **un**iversal-**in**dividual cross-at**tention** (Unitention), an end-to-end trainable module explicitly facilitating that sort of fusion. Thorough experiments verify Unitention's validity on both basic and advanced backbones, *e.g.*, improving Swin and VAN by $0.5 - 1.0\%$ in ImageNet validation accuracy without modifying the backbone. More surprisingly, $k$-NN classifiers with Unitention can even surpass the trained linear classifiers, and further demonstrating the superiority of learned representations. More results show that Unitention can generalize to other modalities and fine-grained tasks. We hope Unitention will serve as a generic module which can improve even the most powerful deep models to date, and could inspire more future works towards better utilization of dataset-level information.

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

## A   More details on the cross-attention module in Unitention

Table 5: **Hyperparameters of cross-attention block** in our Unitention architecture.

| Size | Hidden Dim | # Heads | MLP Ratio | Drop Path | Layer Scale |
|------|-----------|---------|-----------|-----------|-------------|
| Small | 384 | 8 | 3.0 | 0.08 | 1.0 |
| Base | 768 | 12 | 4.0 | 0.1 | 1.0 |
| Large | 1024 | 16 | 4.0 | 0.2 | 0.1 |

We propose three levels of Unitention, from small to large, as in table 5. In practice, RegNetY-16GF uses the Large block, and ResNet-152 and ViT-Base use a Base one. All other models use the Small block. The extra parameters and computations introduced by Unitention are detailed below in table 6.

Table 6: **Extra parameters and computations** introduced by Unitention.

| Model | ResNet-50 | ResNet-101 | ResNet-152 | ViT-S | Swin-T |
|-------|-----------|------------|------------|-------|--------|
| Para (M) | 25.6 | 44.5 | 60.2 | 22 | 29 |
| $\Delta$Para (M) | +1.4 | +1.4 | +6.9 | +1.4 | +1.4 |
| FLOPS (G) | 4.1 | 7.9 | 11.6 | 4.6 | 4.5 |
| $\Delta$FLOPS (G) | +0.3 | +0.3 | +1.2 | +0.3 | +0.3 |

We also measure the wall-clock time and memory footprint of ResNet-50 training (300 epochs with 2048 batch size) on 8 Tesla V100s. The results are reported in table 7. Where Unitention introduces about 5% additional cost. Note this can easily be further reduced by operators such as FlashAttention, as Unitention uses a naive single-query cross-attention.

Table 7: **Wall-clock time and memory footprint** measured.

| Model | GPU hours | Peak GPU Mem |
|-------|-----------|--------------|
| ResNet-50 | 242.0 | 13.2 |
| ResNet-50 w/ Unitention | 255.1 | 13.8 |

## B   More details on the inference-only case study

We provide a PyTorch implementation of the case study in section 2.1. Suppose a supervised-trained model has processed all training and validation data samples to get all intermediate features (after the global average pooling of the backbone) and logits1 (after the fully-connected classifiers).

In this implementation, top-1 accuracies of "U. Acc.", "I. Acc.", and "U-I. Acc." (defined in section 2.1) are calculated and returned. The results are visualized in figure 2.

```python
@torch.no_grad()
def evaluate(t_fea, t_tar, v_fea, v_lg1, v_tar, K=20):
    """
    Args:
        t_fea (Nt x D): features of training data
        t_tar (Nt x 1): targets of training data
        v_fea (Nv x D): features of validation data
        v_lg1 (Nv x #CLS): logits1 of validation data
        v_tar (Nv x 1): targets of validation data
    Returns:
        three accuracies
```

```python
    """
    t_fea, v_fea = F.normalize(t_fea, dim=1, p=2), F.normalize(v_fea,
        dim=1, p=2)

    num_classes = 1000
    i_correct = u_correct = 0
    ui_correct = total_preds = 0

    t_fea = t_fea.t()
    retrieval_one_hot = torch.zeros(K, num_classes).to(t_fea.device)

    for fea, logits1, tar in zip(
            v_fea.chunk(100),
            v_lg1.chunk(100),
            v_tar.chunk(100),
        ):
        bs = tar.shape[0]
        total_preds += bs

        # calculate the dot product and compute k-nearest neighbors
        sim = torch.mm(fea, t_fea)
        distance, choice = sim.topk(K, largest=True, sorted=True)
        candidates = t_tar.view(1, -1).expand(bs, -1)
        retrieved_neighbors = torch.gather(candidates, 1, choice)

        # accumulate the k-nearest neighbors' similarities
        retrieval_one_hot.resize_(bs * K, num_classes).zero_()
        retrieval_one_hot.scatter_(1, retrieved_neighbors.view(-1, 1), 1)
        distances_transform = distance.clone().div_(5).exp_()
        sims = torch.sum(
           torch.mul(
               retrieval_one_hot.view(bs, -1, num_classes),
               distances_transform.view(bs, -1, 1),
           ),
           1,
        )

        # predict
        tar_t = tar.data.view(-1)
        u_correct += sims.argmax(1, False).eq(tar_t).int().sum().item()
        i_correct += logits1.argmax(1, False).eq(tar_t).int().sum().item()
        ui_correct += (logits1+get_knn_logits(sims)).argmax(1,
            False).eq(tar_t).int().sum().item()

    return (u_correct/total_preds, i_correct/total_preds,
        ui_correct/total_preds)

def get_knn_logits(p):
  p = (p+1).log()
  p = (p-p.mean(1, keepdims=True)) / p.std(1, keepdims=True) / 20
  return p
```

## C  IMAGENET TRAINING CONFIGURATIONS

We refer to the hyperparameters in "ResNet Strikes Back" (Wightman et al., 2021) to conduct ImageNet experiment. These hyperparameters are listed in table 8.

## D  INATURELISTS TRAINING CONFIGURATIONS

Following the convention in Liu et al. (2021), we first load the checkpoints of Swin-Transformers pre-trained on ImageNet, and then fine-tune them on iNaturelist datasets. The hyperparameters are

Table 8: **ImageNet training configurations taken from Wightman et al. (2021).**

| Configuration | Value | Configuration | Value |
|---|---|---|---|
| Image resolution | 224 | Epochs | 300 |
| Test image crop | 0.95 | Batch size | 2048 |
| Optimizer | LAMB | Learning rate | 5e-3 |
| Scheduler | consine | Weight decay | 0.02 |
| Repeated aug. | ✗ | Dropout | ✗ |
| Rand aug. | 7/0.5 | Stoch. depth | ✓ |
| Gradient clip. | ✗ | BCE loss | ✗ |
| Mixup alpha | 0.1 | Label smoothing | 0.0 |
| Cutmix alpha | 1.0 | EMA | ✗ |

basically the same as those in table 8, but a smaller number of training epochs (100), a smaller learning rate (2e-4), and a larger weight decay (0.05) is used.

# E  1D-Signal Classification Training Configurations

We refer to the code implementations of Wightman et al. (2021) and Karim et al. (2017) for our experiments. Details are in table 9. The learning rate scheduler (LR scheduler in the table) is "ReducedOnPlateau" referring to Karim et al. (2017).

Table 9: **Signal classification training hyperparameters.**

| Configuration | Value | Configuration | Value |
|---|---|---|---|
| Epochs | 2000 | Warm-up | 100 |
| Batch size | 128 | Weight decay | 1e-5 |
| Optimizer | Adam | LR Scheduler | Plateau |
| Max LR | 1e-3 | Min LR | 1e-5 |
| Gradient clip. | ✗ | BCE loss | ✗ |
| Label smoothing | 0.05 | EMA | 0.99 |

# F  Detection and Segmentation Results

We tested Unitention on object detection (MS-COCO 2017 Lin et al. (2014)), instance segmentation (MS-COCO 2017 Lin et al. (2014)), and semantic segmentation (ADE20k Zhou et al. (2017)) tasks. The results are as follows:

Table 10: **COCO object detection and instance segmentation.**

| Object Detection | Image Resize | Schedule | $AP^{box}$ | $AP^{box}_{50}$ | $AP^{box}_{75}$ |
|---|---|---|---|---|---|
| ResNet-50 | (384, 600) | 1× | 38.9 | 59.6 | 42.7 |
| ResNet-50 w/ Unitention | (384, 600) | 1× | 40.1 (+1.2) | 59.9 (+0.3) | 44.2 (+1.5) |

| Instance Segmentation | Image Resize | Schedule | $AP^{mask}$ | $AP^{mask}_{50}$ | $AP^{mask}_{75}$ |
|---|---|---|---|---|---|
| ResNet-50 | (384, 600) | 1× | 35.4 | 56.5 | 38.1 |
| ResNet-50 w/ Unitention | (384, 600) | 1× | 36.3 (+0.9) | 57.1 (+0.6) | 39.0 (+0.9) |

Table 11: **ADE20k semantic segmentation.**

| Semantic Segmentation | Crop Size | Schedule | mIoU |
|---|---|---|---|
| Swin-T | (512, 512) | 160k | 44.5 |
| Swin-T w/ Unitention | (512, 512) | 160k | 45.2 (+0.7) |

The improvements of Unitention are non-trivial among all these dense prediction tasks. These cross-task improvements shown in table 10 and table 11, along with the cross-modal results in the paper table 1 and table 3, have provided a comprehensive and robust validation of Unitention.

## G   VISUALIZATION ON LEARNED REPRESENTATIONS

To examine the linear separability of deep features, we consider using the *linear* principal component analysis (PCA) to project all high-dimensional feature vectors to 2D, instead of the t-distributed stochastic neighbor embedding (t-SNE). We randomly choose ImageNet class indices of $[331, 117, 153, 362, 471, 333, 275, 112, 457, 349]$ for this visualization. From figure 4 one can see that the deep features of Unitention are more separable than baseline features. This is consistent with our observation of Unitention's excellent $k$-NN performance (in table 1) and again demonstrates the superiority of its deep features.

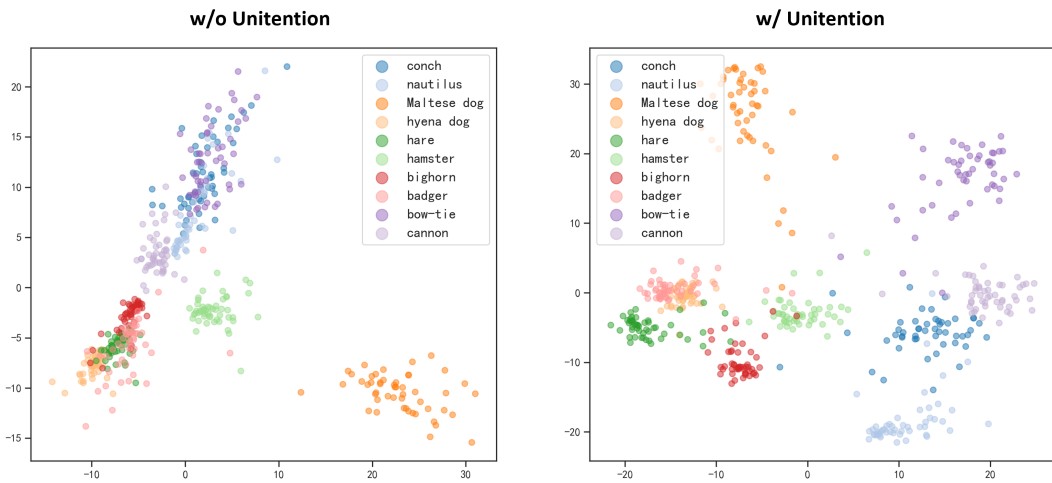

Figure 4: **PCA visualization** of classes as represented after backbone model (left) or the model with Unitention (right). 10 classes on ImageNet test set are randomly selected for visualization. For each class, all of the 50 test images are used. X-axis and y-axis correspond to the first two principal components. It can be seen that the deep features of Unitention are more separable.

