# OpenReview forum: "Unitention: Attend a sample to the dataset"
_ICLR.cc/2024/Conference — Submitted to ICLR 2024_

### Official Review · Reviewer_Ubis · 2023-10-29

**Soundness:** 2 fair
**Presentation:** 3 good
**Contribution:** 3 good
**Rating:** 5
**Confidence:** 5

**Summary:**

This paper proposes a trainable module termed Unitention, an abbreviation for universal-individual cross-attention, to improve deep features of a given neural network by attending the features of a data sample to those of the entire dataset. They are inspired by traditional visual encoding methods, such as bag-of-visual-words, to attend to the entire datasets for visual recognition. The paper validates the effectiveness of this new approach through certain experimental evaluations.

**Strengths:**

1. The paper proposes an attention mechanism to attend to the entire dataset rather than to individual images for visual recognition,  which is a kind of novel approach.
2. The paper is well-written with good organization.
3. The paper justifies its claim through several good experimental settings. For instance, they performed a training-free study on the attention mechanism to justify their design.
4. They perform detailed experimental evaluation, and the effectiveness of this approach is validated by the improvement of visual recognition performance.

**Weaknesses:**

While the methods approach the visual recognition problem through a novel perspective, i.e., the contextual information of the dataset, I suspect this is not very fair for the testing scenario, as testing should be performed via individual images, one by one. If one intends to include the contextual information from datasets, one also should not include the extra data in the dataset during the testing phase, otherwise, it is not fair.
I only have this concern.

**Questions:**

No other questions, but how is including extra data during the testing phase fair for evaluation?

---

> ### Author Response · Authors · 2023-11-21
> **Thank you for your comprehensive summary and useful comments! Reply to Reviewer Ubis**
>
> Thank you very much for your strong support and the insightful comments. We appreciate for the comments on "through a novel perspective", "good organization" and "justifies its claim through several good experimental settings". We would like to address your concerns as follows.
>
> > I suspect this is not very fair for the testing scenario, as testing should be performed via individual images, one by one. If one intends to include the contextual information from datasets, one also should not include the extra data in the dataset during the testing phase, otherwise, it is not fair. I only have this concern.
>
> A: We thank the reviewers for raising concerns about improving the fairness of our methodology. Despite this concern, we believe that Unitention's test scenario is still *fair*. The reason for this is as follows:
>
> - 1. It is generally acknowledged that during the testing, the use of training data is fair, whereas the use of test data (e.g., using **multiple** test samples in an inference, **not one by one**) is a data leakage. For instance, many machine learning models like k-NN, Decision Tree, SVM, Gaussian Process, are all based on training data's features. At the time of testing, these models already have information about the entire training dataset. They are considered fair, because no features of test data are memorized during the testing.
>
> - 2. When testing, Unitention behaves quite similarly to these traditional machine learning models. It also uses multiple features from the training data (like what k-NN and Decision Tree do) to gain some understanding at the dataset level. So this is also a fair testing.
>
> - 3. We conclude that the reason why Reviewer Ubis has such concerns is Unitention uses a new inference process that is fundamentally different from mainstream deep learning models: it uses a universal-individual feature fusion mechanism, whereas typical deep models have no such explicit dataset-level (universal) features. This reaffirms the originality and uniqueness of our approach.
>
> We have updated our manuscript to explain more on the fairness of Unitention's testing phase. We hope our replies can address your concerns.
>
> &nbsp;
>
> **Note from the authors**: we have updated several important experiment results, which we think would be valuable and is also suggested by other reviewers: we now tested Unitention on object detection (MS-COCO 2017), instance segmentation (MS-COCO 2017), and semantic segmentation (ADE20k) tasks. The results are as follows:
>
> | Object Detection &nbsp; | Image Resize &nbsp; | Schedule &nbsp; | $\text{AP}^\text{box}$ | $\text{AP}^\text{box}\_{50}$ | $\text{AP}^\text{box}\_{75}$ |
> | --- | --- |  --- |  --- |  --- |  --- |
> | ResNet-50 &nbsp; | (384, 600) | 1$\times$ | 38.9 | 59.6 | 42.7 |
> | ResNet-50 w/ Unitention &nbsp; | (384, 600) | 1$\times$ | 40.1 (+1.2) &nbsp; | 59.9 (+0.3) &nbsp; | 44.2 (+1.5) &nbsp; |
>
> | Instance Segmentation &nbsp; | Image Resize &nbsp;  | Schedule &nbsp; | $\text{AP}^\text{mask}$ | $\text{AP}^\text{mask}\_{50}$ | $\text{AP}^\text{mask}\_{75}$ |
> | --- | --- |  --- |  --- |  --- |  --- |
> | ResNet-50 &nbsp; | (384, 600) | 1$\times$ | 35.4 | 56.5 | 38.1 |
> | ResNet-50 w/ Unitention &nbsp; | (384, 600) | 1$\times$ | 36.3 (+0.9) &nbsp; | 57.1 (+0.6) &nbsp; | 39.0 (+0.9) &nbsp; |
>
> | Semantic Segmentation &nbsp; |  Crop Size &nbsp; | Schedule &nbsp; | $\text{mIoU}$ |
> | --- | --- |  --- |  --- |
> | Swin-T &nbsp; | (512, 512) | 160k | 44.5 |
> | Swin-T w/ Unitention &nbsp; | (512, 512) | 160k | 45.2 (+0.7) |
>
> The improvements of Unitention are non-trivial among all these dense prediction tasks. These cross-task results, along with the cross-modal results in the paper, provide a comprehensive and robust validation of Unitention. We highlight the results here, and you can also check them out in our updated appendix.
>
>
> &nbsp;
>
> ------------
>
> **Last, thanks so much for helping us improve this work through your comprehensive perspective. If you have any additional questions or anything you would like to discuss in more detail, please feel free to let us know. If you find we have successfully addressed your worries, please kindly consider re-evaluating our work, thanks a lot.**

---

### Official Review · Reviewer_CKwM · 2023-10-31

**Soundness:** 3 good
**Presentation:** 2 fair
**Contribution:** 3 good
**Rating:** 6
**Confidence:** 5

**Summary:**

This research introduces a sample-dataset interaction mechanism through cross-attention block, termed Unitention, which contributes to enhance the deep feature for classification. To be specific, the universal distribution for dataset is characterized by a collection of class-specific prototypes, and accumulated in an EMA manner with label annotation. Unitention module is evaluated on multiple architectures.

**Strengths:**

(+) Adopting the holistic representation to depict the dataset distribution is an interesting exploration. And it maybe inspire to subsequent works.

(+) The universal information and individual information is interacted with a simple-yet-effective cross-attention block.

(+) The proposed method is evaluated on extensive architectures, and obtains healthy gains.

**Weaknesses:**

(-) The relation between Unitention and codebook/k-NN is somehow overclaimed.

(1) Typically, codebook is constructed using unsupervised methods, often employing K-means. However, in this study, But in this work, it seems that the codebook is dependent on labels.

(2) Unlike the selection of k-nearest neighbor in k-NN, the unitention module instead aggregates all feature candidates for cross-attention. In fact, it seems that such unitention module does not allow a partial neighbor, i.e., K less than class number, because each candidate responses a specific class.


Although Unitention shares some commonalities with coding algorithm, it is essential to clearly highlight their differences to prevent misleading readers.


(-) The migration ability for annotated universal bank is limited.
The Universal bank essentially serves as a class prototype, requiring label annotations for aggregation. Such mechanism, which relies on annotation information, faces the challenge when migrating to other downstream task with different categories.
Additionally, such character should be clearly outlined in section 2.2. Moreover, it is advisable to movie the updating principle for universal banks is recommended to section 2.2.

(-) The algorithm is not clearly demonstrated.
As the clarification in paper, the training-free approach is like a variation of k-NN. But how does the classifier come from for test phase? By aggregating training samples?

(-) Lack of some empirical studies.

(1) It is recommended to analysis Unitention training/inference efficiency. In particular, the computational and storage demands for the additional 1000xC class-level feature bank.

(2) Lack of the evaluation on other popular vision tasks, e.g., detection or segmentation.

**Questions:**

Other Comments:

Q1: About the hyperparameter analysis on $\tau$. It seems that tuning $\tau$ obtains positive gains with momentum 0.8 setting, i.e. row 7, 9, 10 and 11 in Table 4. Why does fix $\tau$ as 1 finally?

Q2: In Table1, the k-NN accuracy is obtained by the model trained with only k-NN classifier?

Q3: It seems that there is a typo in section 3.2, i.e., BL* in tab:imn -> BL* in tab:1.

---

> ### Author Response · Authors · 2023-11-21
> **Thank you for your support and professional, detailed feedback! Response to reviewer CKwM [1/2]**
>
> Many thanks to Reviewer CKwM's comments and advice, which help us a lot to improve our work. We are also grateful to Reviewer CKwM for evaluating our work as "an interesting exploration", "simple-yet-effective", and perhaps inspiring subsequent works. See below for the answers to your questions and comments.
>
> ------------------
>
> > (W1) Although Unitention shares some commonalities with coding algorithm, it is essential to clearly highlight their differences to prevent misleading readers. (1): traditional codebook are unsupervised, while codebook here is dependent on labels
>
> A: We highly appreciate the helpful advice to avoid misleading readers. Traditional codebook motivates our high-level idea of using some universal "bank" to complement the deep feature. This high-level idea of Unitention does not limit the specific bank structure to be implemented. As mentioned in Section 2.2, there are a lot available ways to implement the bank, e.g., **label-free** FIFO queue, or **label-based** average class centers. In practice, we choose the class center only for its performance, and this does not contradict our original high-level motivation. We have updated the statements to mitigate any potential misunderstanding.
>
> > (W1) (2): unlike the selection of k-nearest neighbor in k-NN, the unitention module instead aggregates all feature candidates ...
>
> A: Thanks for the professional comment. When we design the Unitention module, we don't try to *imitate*, or *copy* a k-NN algorithm, but rather to borrow the *design philosophy* behind k-NN to produce a "universal" understanding of the dataset. As you pointed out, k-NN is performed like a "hard" nearest neighbour selection and is therefore hard to optimize. So we consider using the "soft" version of neighbor aggregation (softmax attention) as a trainable module for our quest for universal understanding. In other words, k-NN builds the universal understanding, but is not the only way to do so.
>
> &nbsp;
>
> > (W2) unitention faces the challenge when migrating to other downstream task with different categories. Additionally, such character should be clearly outlined in section 2.2. Moreover, it is advisable to movie (move) the updating principle for universal banks to section 2.2.
>
> A: Many thanks to the valuable advice. Besides adjusting the according statements in our manuscript, we further clarify about the **migration ability** of Unitention.
>
> - 1. Unitention is not a pretraining method, and it does not aim to give some pretrained backbones. We tend to think of Unitention as an end application: if someone wants to tackle the classification task on domain B, with a backbone pretrained on domain A, we would suggest to load an existing pretrained backbone (without Unitention), and train it with Unitention directly on domain B (so that the class prototypes are built from scratch on B). This is what we actually do on iNaturalist benchmark, as introduced in Section 3.3 and Appendix D. We will refine Section 3.3 and Appendix D to make this clearer.
>
> - 2. If there is a real need to provide migration capabilities for Unitention, we could use a FIFO queue bank to achieve this. However, this may not perform as well as class centers. This can be explained by the "no free lunch" theorem: generality often comes at the expense of performance.
>
> &nbsp;
>
> > (W3): the training-free approach is like a variation of k-NN. But how does the classifier come from for test phase? By aggregating training samples?
>
> A: We first train a classification model using a standard supervised learning configuration. To obtain a k-NN classifier, we remove the linear classification head of the model, to get a pure backbone $f(\cdot)$.The deep features $f(x\_i)$ of all training samples $x\_i$ are used to build the k-NN classifier.
>
> &nbsp;
>
> > (W4) Lack of some empirical studies. (1): training/inference efficiency
>
> A: We take ResNet-50 training (300 epochs with 2048 batch size) on 8 Tesla V100s as an instance. Unitention adds around 5% extra overhead. Note this can be further optimized by operators such as FlashAttention, as Unitention performs a naive single-query cross-attention:
>
> | Model | GPU hours &nbsp; | Peak GPU Mem |
> | --- | --- | --- |
> | ResNet-50 &nbsp; | 242.0 | 13.2 |
> | ResNet-50 w/ Unitention &nbsp; | 255.1 | 13.8 |
>
> (to be continued)

---

> ### Author Response · Authors · 2023-11-21
> **Thank you for your support and professional, detailed feedback! Response to reviewer CKwM [2/2]**
>
> > (W4) (2): Lack of the evaluation on other popular vision tasks, e.g., detection or segmentation.
>
> A: We highly agree (also stated in the old Limitation section of paper) that evaluation on detection or segmentation tasks can make our empirical conclusion more robust. We have done these experiments and reported the performance in the appendix, where we tested Mask R-CNN ResNet50-FPN on MS-COCO, and the UperNet Swin-T on ADE20k. The results are:
>
> | Object Detection &nbsp; | Image Resize &nbsp; | Schedule &nbsp; | $\text{AP}^\text{box}$ | $\text{AP}^\text{box}\_{50}$ | $\text{AP}^\text{box}\_{75}$ |
> | --- | --- |  --- |  --- |  --- |  --- |
> | ResNet-50 &nbsp; | (384, 600) | 1$\times$ | 38.9 | 59.6 | 42.7 |
> | ResNet-50 w/ Unitention &nbsp; | (384, 600) | 1$\times$ | 40.1 (+1.2) &nbsp; | 59.9 (+0.3) &nbsp; | 44.2 (+1.5) &nbsp; |
>
> | Instance Segmentation &nbsp; | Image Resize &nbsp;  | Schedule &nbsp; | $\text{AP}^\text{mask}$ | $\text{AP}^\text{mask}\_{50}$ | $\text{AP}^\text{mask}\_{75}$ |
> | --- | --- |  --- |  --- |  --- |  --- |
> | ResNet-50 &nbsp; | (384, 600) | 1$\times$ | 35.4 | 56.5 | 38.1 |
> | ResNet-50 w/ Unitention &nbsp; | (384, 600) | 1$\times$ | 36.3 (+0.9) &nbsp; | 57.1 (+0.6) &nbsp; | 39.0 (+0.9) &nbsp; |
>
> | Semantic Segmentation &nbsp; |  Crop Size &nbsp; | Schedule &nbsp; | $\text{mIoU}$ |
> | --- | --- |  --- |  --- |
> | Swin-T &nbsp; | (512, 512) | 160k | 44.5 |
> | Swin-T w/ Unitention &nbsp; | (512, 512) | 160k | 45.2 (+0.7) |
>
> Unitention brings similar gains across these three dense prediction tasks, which showcases its effectiveness.
>
> &nbsp;
>
> > (Q1) It seems that tuning $\tau$ obtains positive gains with momentum 0.8 setting, i.e. row 7, 9, 10 and 11 in Table 4. Why does fix $\tau$ as 1 finally?
>
> A: Thanks for pointing out the typo. In row 9, 10, and 11 in Table 4, $\tau$ should be 1.0. We have corrected this.
>
> > (Q2) In Table1, the k-NN accuracy is obtained by the model trained with only k-NN classifier?
>
> A: As explained earlier in the response to (W3), we first train the classification model using standard supervised learning. We then  remove the learned linear classification head, and perform k-NN inference on the pure backbone. We have updated the according statements in Section 2.1.
>
> > (Q3) It seems that there is a typo in section 3.2, i.e., BL* in tab:imn -> BL* in tab:1.
>
> A: We have fixed this. Many thanks for your thorough proofreading.
>
>
> &nbsp;
>
> --------------------
>
> **We sincerely appreciate the encouragement and professional, detailed advice from reviewer CKwM. We took all your suggestions into account and updated our manuscript. Please let us know if you have any further questions. If you feel that the above improvements help clear up your doubts and further improve our work, you can kindly consider a re-evaluation, thank you!**

---

### Official Review · Reviewer_yV5u · 2023-10-31

**Soundness:** 2 fair
**Presentation:** 3 good
**Contribution:** 2 fair
**Rating:** 5
**Confidence:** 3

**Summary:**

The paper proposes an attention module named Unitention in order to improve the performance of different backbone models with considering the universal information from entire dataset. Unitention takes the output feature vector of a backbone as input and enhances the feature with universal feature from a proposed universal feature bank using cross-attention. The universal feature bank is updated using a momentum mechanism according the classes of input samples. The results show that Unitention demonstrates performance gains on different backbone models and can generalize to multiple modalities.

**Strengths:**

(1) The proposed method is simple and logical,

(2) Sufficient experiments with multiple backbones and different modalities are provided.

(3) This paper is well-written and easy to follow.

**Weaknesses:**

(1) Parameters, flops and inference speeds are not provided. It shows that an Unitention module contains multiple FC layers and a feature bank. When applying an attention module to a backbone, it will improve its performance but also increase its parameter and flop usually. The authors should provide the parameters, flops and inference speeds for models with and without Unitention module, so that we can make sure whether Unitention can make a good trade-off between performance and cost. For example, the parameters, flops and inference speeds of ResNet50, ResNet101, ResNet152, ViT-Small, Swin-Tiny with and without Unitention.

(2) The authors considered different models and modalities in the experiments. While Unitention was only tested in single-label global-feature-based classification tasks in the experiments. I think the authors could provide some experimental results of Unitention for different tasks such as multi-label classification to show its generalization ability.

(3) From my point of view, Unitention is an enhanced design of classification head focusing on the information of classes, since it uses feature vectors after the global average pooling operation. I think the authors should compare Unitention with other classification head designs, for example, iSQRT-COV[1]. Or the authors could show that Unitention is complementary with them, which can still demonstrate performance gains on the models with other heads.

[1] Wang, Qilong, et al. "Deep cnns meet global covariance pooling: Better representation and generalization." IEEE transactions on pattern analysis and machine intelligence 43.8 (2020): 2582-2597.

**Questions:**

(1) In the ablation study, how does Unitention update the feature bank when the number of class centers is less than number of classes?

(2) I'm wondering about the performance of Unitention when the feature bank is replaced by trainable parameters. That is, the parameters in the feature bank is updated according to the gradient but not proposed method.

(3) Will the backbones with Unitention still show performance gains when fine-tuned on the downstream tasks such as detection and segmentation?

---

> ### Author Response · Authors · 2023-11-21
> **Thank you for the thorough and creative comments! Response to reviewer yV5u [1/2]**
>
> We thank Reviewer yV5u for their thorough responses and comprehensive suggestions. We appreciate that the effectiveness and simplicity of Unitention is acknowledged. Below we list our responses to each of the comments:
>
> ----------------
>
> > (W1) Parameters, flops and inference speeds are not provided.
>
> A: We have reported these in our updated version, hoping they can help the readers better see the efficiency-effectiveness trade-off in our Unitention. For instance, the increases in parameters and flops are:
>
> | Model | ResNet-50 | ResNet-101 | ResNet-152 | ViT-S &nbsp; | Swin-T &nbsp; |
> | --- | --- | --- | --- | --- | --- |
> | Para (M) &nbsp;&nbsp; | 25.6 | 44.5 | 60.2 | 22 | 29 |
> | $\Delta$Para (M) &nbsp;&nbsp; | +1.4 | +1.4 | +6.9 | +1.4 | +1.4 |
> | FLOPS (G) &nbsp;&nbsp; | 4.1 | 7.9 | 11.6 | 4.6 | 4.5 |
> | $\Delta$FLOPS (G) &nbsp;&nbsp; | +0.3 | +0.3 | +1.2 | +0.3 | +0.3 |
>
> Overall, we believe the extra overhead is theoretically acceptable. We also measure the wall-clock time and memory footprint of ResNet-50 training (300 epochs with 2048 batch size) on 8 Tesla V100s:
>
> | Model | GPU hours &nbsp; | Peak GPU Mem |
> | --- | --- | --- |
> | ResNet-50 &nbsp; | 242.0 | 13.2 |
> | ResNet-50 w/ Unitention &nbsp; | 255.1 | 13.8 |
>
> Where Unitention introduces about 5% additional cost. Note this can easily be further reduced by operators such as FlashAttention, as Unitention  uses a naive single-query cross-attention.
>
> &nbsp;
>
> > (W2) More experiments besides single-label global-feature-based classification tasks to show the generalization ability.
>
> A: We highly agree to benchmark Unitention on more challenging tasks beyond the single-and-global classification, e.g., the per-pixel-classification (semantic segmentation). As we also expressed in the Limitation section, we would like to test Unitention on dense prediction tasks including detection and segmentation. Now we've finished these and reported the results in appendix, where we tested a CNN (Mask R-CNN ResNet50-FPN) on MS-COCO, as well as a transformer (UperNet Swin-T) on ADE20k:
>
> | Object Detection &nbsp; | Image Resize &nbsp; | Schedule &nbsp; | $\text{AP}^\text{box}$ | $\text{AP}^\text{box}\_{50}$ | $\text{AP}^\text{box}\_{75}$ |
> | --- | --- |  --- |  --- |  --- |  --- |
> | ResNet-50 &nbsp; | (384, 600) | 1$\times$ | 38.9 | 59.6 | 42.7 |
> | ResNet-50 w/ Unitention &nbsp; | (384, 600) | 1$\times$ | 40.1 (+1.2) &nbsp; | 59.9 (+0.3) &nbsp; | 44.2 (+1.5) &nbsp; |
>
> | Instance Segmentation &nbsp; | Image Resize &nbsp;  | Schedule &nbsp; | $\text{AP}^\text{mask}$ | $\text{AP}^\text{mask}\_{50}$ | $\text{AP}^\text{mask}\_{75}$ |
> | --- | --- |  --- |  --- |  --- |  --- |
> | ResNet-50 &nbsp; | (384, 600) | 1$\times$ | 35.4 | 56.5 | 38.1 |
> | ResNet-50 w/ Unitention &nbsp; | (384, 600) | 1$\times$ | 36.3 (+0.9) &nbsp; | 57.1 (+0.6) &nbsp; | 39.0 (+0.9) &nbsp; |
>
> | Semantic Segmentation &nbsp; |  Crop Size &nbsp; | Schedule &nbsp; | $\text{mIoU}$ |
> | --- | --- |  --- |  --- |
> | Swin-T &nbsp; | (512, 512) | 160k | 44.5 |
> | Swin-T w/ Unitention &nbsp; | (512, 512) | 160k | 45.2 (+0.7) |
>
> The consistent performance gains from Unitention showcase its generalization ability.
>
> &nbsp;
>
> > (W3) the authors could show that Unitention is complementary with other classification head designs, for example, iSQRT-COV [r1]
>
> A: Thanks for this valuable advice. Since Unitention offers a way to improve heads that is orthogonal to existing methods (i.e. through a cross-data individual-universal fusion mechanism), it is worth investigating whether it can complement those existing designs. In addition to iSQRT-COV [r1] on ResNet, we also test another different head design (SoT [a1] on Vision Transformers, suggested by reviewer zCBo) to make the results more convincing. The results are as follows and are also included in the appendix:
>
> | Model | baseline (our impl.) &nbsp; | w/ [r1,a1] (our impl.) &nbsp; | w/ [r1,a1] and Unitention |
> | --- | --- | --- | --- |
> | ResNet-50 &nbsp; | 78.9 | 80.1 (w/ [r1]) | 80.8 (w/ [r1] and Unitention) |
> | Swin-T | 81.3 | 82.6 (w/ [a1]) | 83.2 (w/ [a1] and Unitention) |
>
> From the results one can see that Unitention can further improve the performance of other head designs due to its new universal-individual mechanism.
>
> &nbsp;
>
> (to be continued)

---

> ### Author Response · Authors · 2023-11-21
> **Thank you for the thorough and creative comments! Response to reviewer yV5u [2/2]**
>
> > (Q1) how does Unitention update the feature bank when the number of class centers is less than number of classes?
>
> A: At the begging of training, we random select 100/1000 classes and only maintain the class centers of them during the whole training. We have updated this detail in ablation study.
>
> > (Q2) performance of trainable feature bank?
>
> A: Thanks for such a creative ablation. Following the setting of Section 3.5, we tried it out and found it achieved an accuracy of 79.4%, close to a FIFO queue. We feel that a trainable feature bank would still be hopeless to beat class centers, as class centers directly use the information from labels. This ablation is added to our study and we hope it can help the readers better understand our Unitention.
>
> > (Q3) downstream performance of detection and segmentation?
>
> A: See the response to (W2) above.
>
>
> &nbsp;
>
> **Again, thank you for your insightful and comprehensive advice that greatly improved our Unitention. We have carried out the suggested experiments and made every effort to address the concerns and weaknesses. If there are any further questions or concerns, please let us know. If you feel that we have successfully addressed your concerns, please kindly consider a re-evaluation. Thank you.**
>
> --------------
>
> [r1] Wang, Qilong, et al. Deep cnns meet global covariance pooling: Better representation and generalization. T-PAMI 2020.
>
> [a1] Jiangtao Xie, et al. Sot: Delving deeper into classification head for transformer. arXiv preprint arXiv:2104.10935, 2021.

---

### Official Review · Reviewer_zCBo · 2023-11-01

**Soundness:** 3 good
**Presentation:** 2 fair
**Contribution:** 2 fair
**Rating:** 5
**Confidence:** 4

**Summary:**

This paper attempts to improve the performance of a deep neural network (DNN) by modifying its head structure instead of the backbone structure. Specifically, the authors present a trainable fusion module called Unitention. The basic idea of Unitention is to combine individual feature encoding and universal feature encoding. Specifically, individual feature encoding is just the feature output from a given DNN backbone, while universal feature encoding takes the feature output from this given DNN backbone as the input, and uses a cross-attention module following the self-attention concept in popular transforms to capture sample-to-dataset relations. Then, the output of Unitention is just the sum of the outputs from both individual feature encoding and universal feature encoding. Experiments conducted on image classification (with ImageNet-1K dataset), fine-grained classification (with iNaturalist 2018) and one-dimensional signal classification (with three datasets for device/sensor/medical signals) tasks are provided to show the efficacy of the proposed method.

**Strengths:**

+ The paper is well written in most parts.

+ The idea of the proposed method is easy to understand.

+ Comparative experiments are performed on three types of benchmarks with different deep neural network architectures including convnets and vision transformers.

+ The proposed method shows improvement to baselines on different datasets.

+ The limitations of the proposed method are also discussed.

**Weaknesses:**

- The motivation, the method and related works.

The motivation of this paper is to improve the performance of a deep neural network (DNN) by designing a new head structure, or say enhancing feature encoding for the classification head.

The authors present Unitention, a trainable fusion module that can be inserted after the backbone structure. The basic idea of Unitention is to combine individual feature encoding and universal feature encoding. Specifically, individual feature encoding is just the feature output from a given DNN backbone, while universal feature encoding takes the feature output from this given DNN backbone as the input, and uses a cross-attention module following the self-attention concept in popular transforms to capture sample-to-dataset relations. Then, the output of Unitention is just the sum of the outputs from both individual feature encoding and universal feature encoding.

However, the motivation, the idea and the cross-attention design of the proposed Unitention are not new. As a fundamental research topic, there already exist a large number of previous research works that focus on designing a better head structure or enhancing feature encoding for the classification head. Unfortunately, the authors totally ignore this line of research. In what follows, I just list some representative works as well as recent works in this field.

[1] Mircea Cimpoi, et al. Deep Filter Banks for Texture Recognition and Segmentation. CVPR 2015.

[2] Relja Arandjelovic, et al. NetVLAD: CNN architecture for weakly supervised place recognition. CVPR 2016.

[3] Yang Gao, et al. Compact Bilinear Pooling. CVPR 2016.

[4] Feng Zhu, et al. Learning Spatial Regularization with Image-level Supervisions for Multi-label Image Classification. CVPR 2017.

[5] Mengran Gou, et al. MoNet: Moments Embedding Network. CVPR 2018.

[6] Shilong Liu, et al. Query2label: A simple transformer way to multi-label classification. arXiv preprint arXiv:2107.10834, 2021.

[7] Jiangtao Xie, et al. Sot: Delving deeper into classification head for transformer. arXiv preprint arXiv:2104.10935, 2021.

[8] Ke Zhu, et al. Residual Attention: A Simple but Effective Method for Multi-Label Recognition. CVPR 2021.

[9] Chao Li, et al. NOAH: A New Head Structure To Improve Deep Neural Networks For Image Classification. ICLR 2023 Open Review.

- The experiments.

As I mentioned above, there already exist a large number of previous research works that focus on designing a better head structure or enhancing feature encoding for the classification head. However, the authors totally ignore them throughout the paper, including in the experimental comparison. Comprehensive comparisons of the proposed method to existing works are necessary.

How about the extra training cost of the proposed method against the baseline?

I would like to the transferring ability of Unitention to downstream tasks as they are more important in real applications.

**Questions:**

Please refer to my detailed comments in "Weaknesses" for details.

---

> ### Author Response · Authors · 2023-11-21
> **Thanks for the constructive feedback! But there could be some misunderstandings...  Reply to reviewer zCBo [1/2]**
>
> We appreciate Reviewer zCBo’s constructive feedback, especially the additional related work which is very helpful in highlighting the uniqueness and novelty of our Unitention. However, the judgement of *“the motivation, the idea and the cross-attention design of the proposed Unitention are not new”* seems to misunderstand the core difference between Unitention and related work [r1-r9], thus may lead to a *misjudgment* of the value and novelty of Unitention. We are sorry about this and will clarify this later. We'd like to response to each of concerns as follows.
>
> ----------------
>
> **[To additional related literature required]**
>
> Thanks for pointing out the work **[r1-r9]** which helps a lot to consolidate our work. **[r1-r2] are Bag-of-Words** methods which we have already discussed in Section 4.1/4.3, where we've shown their fundamental difference to Unitention. **For head-enhancing work [r3-r9]**, we did a thorough review and summarized it in a new Related Work subsection in the revised pdf. Now we're addressing the *core concern* on the novelty of Unitention compared to [r3-r9], as follows.
>
> &nbsp;
>
> **[To concerns on novelty]: “given [r1-r9], the motivation, the idea and the cross-attention design of the proposed Unitention are not new”**
>
> (i) &nbsp;The "not new" is somewhat excessive and could be a misjudgment. First, [r3-r5, r7-r9] **didn't use either** a cross-attention operator **or** a cross-data fusion (universal-individual) mechanism.
>
> (ii) &nbsp;Although [r6] uses cross-attention operator (the same as in DETR [a1] and Seq2Seq Transformers [a2, a3]), it **does not use** cross-data fusion mechanism at all (which is the core design of Unitention), thus is still fundamental different from Unitention, as explained at the bottom of page 4. We repeat here that 1) the [r6, a1-a3] uses Q, K, V from one *single* data point and model parameters, whereas 2) Unitention uses key-value pairs from *multiple different* data points. This difference makes cross-attention operator act as two **radically different functions** in [r6, a1-a3] vs. in Unitention: one to select different regions of a *single* image for different recognition targets (one cat's face, two cat's claws, etc.), the other to compare one *single* data with all other data in the *whole* training set and make some *holistic* understanding.
>
> (iii) &nbsp;A full comparison between Unitention and [existing r3-r9, extra ones we surveyed a1-a5] is summarized as follows:
>
> | Methodology | What features to fuse | Cross-attention used? | Cross-data fusion? |
> | --- | --- | --- | --- |
> | Bilinear pooling and improvements [r3, r5] | the feature of one single data point, form one or two models. | $\times$ | $\times$ |
> | Improved global pooling [r7, r8, a5] | the feature of one single data point (r7: different token embedding) | $\times$ | $\times$ |
> | Spatial attention [r4, a4] | the feature of one single data point | $\times$ | $\times$ |
> | Channel-split attention [r9] | non-overlapping feature groups of one feature from one data point | $\times$ | $\times$ |
> | Query-based transformer decoder (DETR, translation model, etc.) [r6, a1-a3] | the feature from one data point | $\checkmark$ | $\times$ |
> | Unitention (ours) | features from different data points | $\checkmark$ | $\checkmark$ |
>
> (iv) &nbsp;In summary, [r1-r9, a1-a5] can be regarded as additional related work to consolidate our work. They won't devalue the main contribution and novelty of Unitention. *Instead, they would* reinforce the originality and uniqueness of Unitention, which is evident from the comparison above and also acknowledged by Reviewer CKwM as well as Reviewer Ubis. We have updated our Related Work to discuss and explain this, hopefully clearing up any possible misunderstandings.
>
> &nbsp;
>
> (to be continued)

---

> ### Author Response · Authors · 2023-11-21
> **Thanks for the constructive feedback! But there could be some misunderstandings...  Reply to reviewer zCBo [2/2]**
>
> **[To other advice/questions about experiments]**
>
> > More comparisons with existing works [r1-r9]
>
> A: Unitention shows a *cross-data* mechanism to improve classification heads, different from existing *single-data-based* head-enhancing modules like [r3-r7]. We highly agree that it is valuable to verify whether Unitention can complement other methods, and this is also suggested by reviewer yV5u. Here, we apply Unitention on two open-sourced model that also experimented on ImageNet (SoT [r7] and iSQRT-COV [a5]), and evaluate the validation accuracy:
>
> | Model | baseline (our impl.) &nbsp; | w/ [r7,a5] (our impl.) &nbsp; | w/ [r7,a5] and Unitention |
> |---|---|---|---|
> | ResNet-50 | 78.9 | 80.1 (w/ [a5]) | 80.8 (w/ [a5] and Unitention)|
> | Swin-T | 81.3 | 82.6 (w/ [r7]) | 83.2 (w/ [r7] and Unitention)|
>
> The further performance improvements demonstrate that Unitention can complement existing advanced head designs. We have included these results in appendix as extra empirical assessments.
>
> &nbsp;
>
> > How about the extra training cost of the proposed method against the baseline?
>
> A: Take ResNet-50 training (300 epochs with 2048 batch size) on 8 Tesla V100s as an example. Unitention introduces about 5% additional overhead (note there is still room for efficiency improvement, e.g., by using FlashAttention, as Unitention performs a naive single-query cross-attention):
>
> |Model| GPU hours &nbsp; |Peak GPU Mem|
> |---|---|---|
> |ResNet-50|242.0|13.2|
> |ResNet-50 w/ Unitention &nbsp; |255.1|13.8|
>
> &nbsp;
>
> > I would like to the transferring ability of Unitention to downstream tasks as they are more important in real applications.
>
> A: We appreciate the suggestion. In the original Limitation section, we left the application of Unitention to object detection, instance segmentation, and semantic segmentation as future work. We have now **completed these three** important downstream evaluations. For the sake of diversity, we test two different families of models (CNNs and Transformers). Specifically, the Mask R-CNN ResNet50-FPN on MS-COCO, and the UperNet Swin-T on ADE20k:
>
> |Object Detection &nbsp; |Image Resize &nbsp; |Schedule &nbsp; | $\text{AP}^\text{box}$ | $\text{AP}^\text{box}\_{50}$ | $\text{AP}^\text{box}\_{75}$ |
> |---|---|---|---|---|---|
> |ResNet-50|(384, 600)|1$\times$|38.9|59.6|42.7|
> |ResNet-50 w/ Unitention &nbsp; |(384, 600)|1$\times$|40.1 (+1.2) &nbsp; |59.9 (+0.3) &nbsp; |44.2 (+1.5) &nbsp; |
>
> | Instance Segmentation &nbsp; | Image Resize &nbsp;  | Schedule &nbsp; | $\text{AP}^\text{mask}$ | $\text{AP}^\text{mask}\_{50}$ | $\text{AP}^\text{mask}\_{75}$ |
> |---|---|---|---|---|---|
> | ResNet-50 | (384, 600) | 1$\times$ | 35.4 | 56.5 | 38.1 |
> | ResNet-50 w/ Unitention &nbsp; | (384, 600) | 1$\times$ | 36.3 (+0.9) &nbsp; | 57.1 (+0.6) &nbsp; | 39.0 (+0.9) &nbsp; |
>
> | Semantic Segmentation &nbsp; |  Crop Size &nbsp; | Schedule &nbsp; | $\text{mIoU}$ |
> |---|---|---|---|
> | Swin-T | (512, 512) | 160k | 44.5 |
> | Swin-T w/ Unitention &nbsp; | (512, 512) | 160k | 45.2 (+0.7) |
>
> The consistent performance gains show that Unitention can boost the performance of downstream dense prediction tasks, further verifying its effectiveness. These results are attached to appendix too.
>
> &nbsp;
>
> **Thanks again for the valuable comments which makes our Related Work more comprehensive and our experiments more solid and robust. We've elaborated on the suggested experiments, and tried every effort to address your concerns on our methodology and experiments. We hope you could kindly consider a re-evaluation if you find our responses are effective. Thank you.**
>
> ----
>
> [r1] Mircea Cimpoi, et al. Deep Filter Banks for Texture Recognition and Segmentation. CVPR 2015.
>
> [r2] Relja Arandjelovic, et al. NetVLAD: CNN architecture for weakly supervised place recognition. CVPR 2016.
>
> [r3] Yang Gao, et al. Compact Bilinear Pooling. CVPR 2016.
>
> [r4] Feng Zhu, et al. Learning Spatial Regularization with Image-level Supervisions for Multi-label Image Classification. CVPR 2017.
>
> [r5] Mengran Gou, et al. MoNet: Moments Embedding Network. CVPR 2018.
>
> [r6] Shilong Liu, et al. Query2label: A simple transformer way to multi-label classification. arXiv preprint 2021.
>
> [r7] Jiangtao Xie, et al. Sot: Delving deeper into classification head for transformer. arXiv preprint 2021.
>
> [r8] Ke Zhu, et al. Residual Attention: A Simple but Effective Method for Multi-Label Recognition. CVPR 2021.
>
> [r9] Chao Li, et al. NOAH: A New Head Structure To Improve Deep Neural Networks For Image Classification. ICLR 2023.
>
> ----
>
> [a1] Carion, Nicolas, et al. End-to-end object detection with transformers. ECCV 2020.
>
> [a2] Vaswani, et al. Attention is all you need. NIPS 2017.
>
> [a3] Radford, et al. Improving language understanding by generative pre-training. 2018.
>
> [a4] Woo, et al. Cbam: Convolutional block attention module. ECCV 2018.
>
> [a5] Wang, Qilong, et al. Deep cnns meet global covariance pooling: Better representation and generalization. T-PAMI 2020.

---

### Author Response · Authors · 2023-11-21
**[Author Rebuttal Summary] on Nov. 21, 2023**

Dear reviewers and chairs,

Thank you all sincerely for your valuable feedback and engagement! Your comments on our work are professional, comprehensive, and will help our work be stronger.

We greatly appreciate that our contributions have been widely recognized by reviewers, as:

- **[originality and simplicity]**:"&nbsp; a novel perspective" (Reviewer Ubis); "interesting exploration; maybe inspire to subsequent work" (Reviewer CKwM); "simple-yet-effective"  (Reviewer Ubis); "simple and logical" (Reviewer yV5u).

- **[good justification, comprehensive evaluation]**:"&nbsp; three types of benchmarks with different deep neural network architectures" (Reviewer zCBo); "sufficient experiments with multiple backbones and different modalities" (Reviewer yV5u); "evaluated on extensive architectures" (Reviewer CKwM); "justifies its claim" (Reviewer Ubis).

&nbsp;

------

During the rebuttal, we made every effort to address the reviewers' concerns. We tried our best to:

- perform the **commonly suggested experiments** on the object detection, instance segmentation, and semantic segmentation tasks, and also report the time/memory cost (suggested by Reviewer zCBo, yV5u, and CKwM).
- address Reviewer zCBo's **concers** about insufficient related work and method novelty.
- address Reviewer zCBo and yV5u's **concers** about more experiments involving existing head designs.
- address Reviewer CKwM's **concers** about paper organization, explanation of motivation, and writing errors.
- address Reviewer Ubis's **concers** about methodology fairness.

&nbsp;

-----

Based on these updates, the main revisions in the manuscript are:

- in Section 1 and 4:&nbsp; adding more discussion on related work (mainly following Reviewer zCBo and yV5u's advice)
- in Section 1 and 2:&nbsp; adding more explanation about motivation and differece between Unitention and k-NN, etc (Reviewer CKwM)
- in Section 3.2 and 3.5:&nbsp; fixing some typos (Reviewer CKwM)
- in Section 3.5:&nbsp; adding additional ablation study (Reviewer yV5u)
- in Section 4:&nbsp; adding more discussion on inference fairness (Reviewer Ubis)
- in Appendix:&nbsp; adding additional details of model parameters, flops, and runtime overheads;&nbsp; adding extra empirical evaluations (detection and segmentation; Unitention on other single-data-based head-enhancing modules).

&nbsp;

-------


**We would be very grateful if you could consider the above improvements. Thanks again for everyone's time and effort. We really appreciate all the feedback and look forward to discussing more during the rest phase and in the future. Thank you!**

Best,

Paper 111 authors

---

### Meta-Review · Area_Chair_kvRJ · 2023-12-09

**Metareview:**

This paper focuses on improving the design of classification heads by leveraging the relation of the current sample to the other samples in the dataset. This line of research is timely and exciting as the inference procedure resembles retrieval-augmented scenarios in large-language models. The reviewers have praised the simplicity of the proposed approach as well as the clarity of the presentation. However, reviewers raised several concerns about this work. Amongst them are the integration and comparison of this approach in a much broader set of related work and a slightly unfair evaluation setup, as the inference procedure leverages both a parametric and non-parametric part. In that case, the authors must consider a simple baseline that takes both a k-NN and a feedforward classification model using late fusion. There are no baselines except for the performance of the raw feedforward image classification model. For these reasons, this paper should be rejected. I recommend that the authors consider the feedback from the reviewing process, improve the paper, and submit it to another venue.

**Justification For Why Not Higher Score:**

Research work not set in a more global context, lacking baselines that would leverage similar inputs during inference.

**Justification For Why Not Lower Score:**

N/A

---

### Decision · Program_Chairs · 2024-01-16

Reject